# Real-World Efficacy and Safety of Apixaban vs. Warfarin in Obese Atrial Fibrillation Patients: Propensity Matching Analysis

**DOI:** 10.3390/biomedicines13020490

**Published:** 2025-02-17

**Authors:** Abdulaziz Algethami, Amjad M. A. Ahmed, Husam Ardah, Seham Alsalamah, Ghadah Alhabs, Ghadah Al Fraihi, Shahad Alanazi, Hend Alharbi, Ahmed Aljizeeri

**Affiliations:** 1King Abdulaziz Cardiac Center, King Abdulaziz Medical City, Ministry of National Guard-Health Affairs, Riyadh 11426, Saudi Arabia; amjadhabib87@hotmail.com (A.M.A.A.); jizeeria@ngha.med.sa (A.A.); 2College of Medicine, King Saud bin Abdulaziz University for Health Sciences, Riyadh 11481, Saudi Arabia; alsalamah299@ksau-hs.edu.sa (S.A.); alhabs312@ksau-hs.edu.sa (G.A.); alfraihi313@ksau-hs.edu.sa (G.A.F.); alenazi302@ksau-hs.edu.sa (S.A.); alharbi334@ksau-hs.edu.sa (H.A.); 3King Abdullah International Medical Research Centre, Riyadh 11481, Saudi Arabia; 4Department of Biostatistics and Bioinformatics, King Abdullah International Medical Research Centre, Ministry of National Guard-Health Affairs, Riyadh 11481, Saudi Arabia; ardahhu@mngha.med.sa

**Keywords:** DOAC, anticoagulation, atrial fibrillation, direct oral anticoagulant, high body weight, obesity, novel oral anticoagulant, stroke, venous thromboembolism, warfarin

## Abstract

**Background/Objectives**: The use of direct oral anticoagulants (DOACs) in obese patients is scarcely studied despite having many advantages over warfarin. Consequently, this study aims to assess the real-world safety and effectiveness of apixaban compared to warfarin in treating atrial fibrillation (AF) in obese patients. **Methods**: A retrospective cohort study examined consecutive AF patients with a BMI of ≥ 30 kg/m^2^ treated with apixaban or warfarin. Patients were started on these medications between January 2015 and December 2021. Efficacy outcomes included ischemic stroke and venous thromboembolism (VTE) occurrences, while safety outcomes encompassed bleeding incidents and mortality rates. Outcomes were assessed following propensity score matching. **Results**: We identified 876 patients treated with either apixaban (414) or warfarin (462). Their mean age was 76.9, with a mean CHA_2_DS_2_VASc score of 4.9 ± 1.97. After matching and compared to warfarin, apixaban was correlated with a lower incidence of all-cause mortality (19.7% vs. 33.7%, *p* < 0.001). The incidences of stroke, venous thromboembolism (VTE), and bleeding events were (4.7% vs. 4.7%, *p* = 1.000), (1.0% vs. 2.6%, *p* = 0.107), and (3.9% vs. 6.2%, *p* = 0.139), respectively. Using Cox-regression model, apixaban was associated with lower mortality risk (HR = 0.728, 95% CI: 0.55–0.97; *p* = 0.030) which remained significant after adjusting for the conventional cardiovascular risk factors and BMI values. **Conclusions**: Apixaban is associated with a trend of reduced incidence of thromboembolism among obese patients with atrial fibrillation and significantly lowers all-cause mortality. Despite earlier concerns, the use of apixaban is an effective and safe alternative to warfarin among obese patients with AF.

## 1. Introduction

Atrial fibrillation (AF), a prevalent supraventricular tachyarrhythmia, is responsible for one-third of hospitalizations due to arrhythmic diseases [1]. While hypertensive heart disease, coronary artery disease, and rheumatic heart disease are often found in AF patients, obesity has recently become a pervasive issue. Nearly one in seven people have a body mass index (BMI) exceeding 30 kg/m², placing them at risk for numerous cardiac diseases, including AF [2,3]. A 5-unit increase in BMI leads to an additional risk of AF of up to 29% [4], with obesity being a factor in 20% of AF cases [5]. Obesity is linked with mortality, hospitalization, heart failure, and thromboembolic events.

Most AF patients undergo long-term oral anticoagulation to minimize the threat of ischemic stroke and other embolic events. Recently, direct oral anticoagulants (DOACs) demonstrated equal or superior efficacy to warfarin due to their specific modes of action, including selective factor Xa inhibitors and direct thrombin inhibitors [6]. However, DOAC use is limited or prohibited in certain scenarios such as mechanical mitral valve irregularities, end-stage renal disease [7], and conditions that increase prothrombotic states like obesity. Obesity can influence the clinical pharmacology of anticoagulants, leading to heightened thrombotic risk or bleeding incidents. In 2016, the Scientific and Standardization Committee of the International Society on Thrombosis and Haemostasis (ISTH) suggested avoiding the use of DOACs in obese or overweight patients unless drug levels are monitored [8].

There are insufficient data on the efficacy and safety of DOACs in obese patients. Our research seeks to address this knowledge gap by comparing the clinical efficacy and safety of apixaban and warfarin in obese adults diagnosed with AF.

## 2. Materials and Methods

### 2.1. Study Design and Settings

This retrospective cohort study involved patients with a BMI exceeding 30 kg/m^2^ diagnosed with AF. These patients were started on either apixaban or warfarin to prevent thromboembolic events in both inpatient and outpatient settings. The study took place from January 2015 to December 2021 at King Abdulaziz Medical City, a tertiary care center under the Ministry of National Guard Health Affairs (MNGHA) in Riyadh, Saudi Arabia. The research received approval from the Institutional Review Board at the joint institution, the King Abdullah International Medical Research Center, with approval number RC22R/130/02.

### 2.2. Identification of Study Participants

We included patients over 18 years of age with a BMI exceeding 30 kg/m^2^ who were diagnosed with atrial AF and initiated on apixaban or warfarin. The AF diagnosis was identified according to the International Classification of Diseases, Tenth Revision, and Clinical Modification (ICD-10-CM) codes: I48.0, I48.1, I48.2, and I48.9. The first appointment following medication initiation was deemed the index visit. We excluded patients with mechanical valves, valvular AF, left ventricular clots, or severe heart failure with an ejection fraction below 30%. We used a non-probabilistic consecutive technique to sample the target population. Patients were sorted into categories based on the type of oral anticoagulant prescribed: apixaban or warfarin. They were further stratified into three BMI groups: group 1 (30 to <35 kg/m^2^), group 2 (35 to <40 kg/m^2^), and group 3 (40 kg/m^2^ or greater).

### 2.3. Data Collection and Study Outcomes

We collected demographic, clinical, and outcome data for all patients from electronic medical records, encompassing inpatient, outpatient, and emergency department visits. Information on patient comorbidities, such as chronic kidney disease, stroke history, and cancer, as well as concurrent medications like antiplatelets and non-steroidal anti-inflammatory drugs (NSAIDs), was noted to evaluate their influence on clinical outcomes. All data were obtained from Electronic Health Records, and patients were monitored until their last follow-up in December 2022.

The primary outcomes evaluated were the incidences of stroke, the recurrence of VTE, and all-cause mortality. The secondary outcomes assessed for safety were major or minor bleeding events defined according to the International Society on Thrombosis and Haemostasis (ISTH) criteria, emergency department visits, and transfusions.

### 2.4. Statistical Analysis

Patients were divided into two groups: apixaban and warfarin. The propensity score matching technique was used to adjust for the influence of differences in the patients’ baseline characteristics and comorbidities on the study outcomes. A propensity score for each patient was generated using a logistic regression model by predicting the treatment received based on demographic, clinical, and procedural characteristics. We matched patients on apixaban with warfarin by performing a 1:1 nearest neighbor match with a caliper width of 0.2 of the standard deviation of the logit of the propensity score. Results were presented as mean ± SD or median (interquartile) for continuous variables, and categorical variables were shown as frequencies and percentages. The study groups were compared using chi-square or Fisher’s exact test for categorical variables and Student’s *t*-test or Wilcoxon rank as appropriate before and after matching. Hazard ratios were calculated using Cox’s proportional hazard models. Stepwise models were generated as follows: Model 1: unadjusted; Model 2: adjusted for age and gender; Model 3: adjusted for Model 2 and risk factors (chronic kidney disease, cancer, hypertension, and a previous history of venous thromboembolism) [9]; Model 4A: adjusted for Model 3 and BMI categories; and Model 4B: adjusted for Model 3 and BMI continuous. Time-to-event curves were generated using Kaplan–Meier methods and compared with the log-rank test for different study outcomes including bleeding, mortality, stroke, and VTE. A 2-sided *p*-value of less than 0.05 was the determination of statistical significance in baseline comparisons. Statistical analyses were conducted using Statistical Analysis Software (SAS version 9.4, SAS institute, Cary, NC, USA) and a matched package.

## 3. Results

### 3.1. Baseline Characteristics

The study included 876 patients (median age 77.0 (67.0, 85.0) years; a total of 65.6% were females) diagnosed with AF and treated with apixaban or warfarin. The average CHA_2_DS_2_VASc (congestive heart failure; hypertension; age ≥ 75 years (doubled); diabetes mellitus; prior stroke, TIA, or thromboembolism (doubled); vascular disease; age 65 to 74 years; and sex category) score was 4.9 ± 1.97; the median BMI was 33.3 (31.0, 37.5) kg/m^2^. The majority (62.1%) had a BMI ranging from 30 to <35, while 21.5% had a BMI between 35 and <40, and only 16.4% had a BMI over 40 kg/m^2^. Traditional cardiovascular risk factors like hypertension (79.7%), diabetes (66.0%), and dyslipidemia (48.5%) were prevalent in the cohort. Apixaban was administered to 414 (47.3%) patients during the study, while the rest, 462 (46.2%), received warfarin. Table 1 presents the baseline clinical characteristics of the cohort, stratified by the anticoagulant used.

We utilized a 1:1 propensity matching to adjust for discrepancies between the two groups, resulting in a final sample of 772 patients. After matching, the median ages between the groups were comparable. The apixaban group consisted of 63.7% females, while the warfarin group constituted 64.8% females. The BMI distribution also showed similarity post-matching; the median BMI was 32.9 (30.9, 36.9) and 33.1 (31.1, 37.6) for the apixaban and warfarin groups, respectively. All factors, including risk factors, past medical history (with the exception of previous cardiac catheterization), and the use of evidence-based medication, showed robust matching with hardly any significant remaining differences between the two groups. The incidence rates of stroke, VTE, and bleeding were found to be 4.7%, 1.8%, and 5.1%, respectively (Table 2).

### 3.2. Safety and Efficacy of Apixaban

After a median follow-up of 1.3 years (interquartile 0.6–2.3 years), 207 deaths were recorded. Of these, 77 out of 386 deaths (19.7%) were in the apixaban group, versus 130 out of 386 patients (33.7%) in the warfarin group. The rate of stroke, VTE, and bleeding was not statistically significant between the groups, as shown in Table 2, although there was a trend to a lower bleeding rate in the apixaban group; the locations of bleeding instances are detailed in Figure 1. Figure 2A presents the Kaplan–Meier survival curves, which show that apixaban was linked to a lower risk of mortality than warfarin, with an unadjusted hazard ratio of 0.728 (95% CI: 0.55–0.97; *p* = 0.030). The mortality risk associated with apixaban is significantly lower even after adjusting for the conventional cardiovascular risk factors and BMI values. (Table 3). When examining different safety outcomes such as stroke, VTE, and bleeding, the Kaplan–Meier survival curves indicated no difference between apixaban and warfarin (Figure 2B–D). Additional figures show the event periodically available in the appendix (Appendix A).

## 4. Discussion

Direct oral anticoagulants offer significant advantages over warfarin, simplifying the choice for many patients. However, they are not as beneficial for a considerable number of AF patients, particularly those with a high BMI, who face an increased risk of adverse thromboembolic events and are often overlooked in existing research studies. Furthermore, since feasibility and ethics are always an issue for randomized clinical trials, the implementation of statistical methods significantly reduces the impact of existing confounders. The propensity-matching score, as one of the statistical solutions, estimates the relevant clinical effects adjusted for given confounders. Therefore, to overcome the literature and design limitations, we utilized a propensity-matching score to prove the extent of efficacy and safety in different unstudied populations. Our analysis of real-world data revealed that obese AF patients on apixaban had a lower overall incidence of efficacy and safety outcomes compared to those on warfarin. Notably, these patients demonstrated significantly reduced all-cause mortality rates, and this trend persisted even after propensity matching and adjusting for known confounders.

### 4.1. The VTE and Stroke Literature on Apixaban vs. Warfarin in Obesity

Our results align with those of Perales et al., who found similar rates of stroke, VTE recurrence, and mortality in morbidly obese patients using rivaroxaban and warfarin [10]. Similar findings were reported by Coons et al., who found no significant variance in VTE recurrence between apixaban and warfarin in a comprehensive clinical study [11]. In a study using an ICD code for morbid obesity, comparable VTE recurrence rates were observed for DOACs and warfarin [12]. Aloi KG (2021), a retrospective analysis of patients with VTE treated in the Veterans Integrated Service Network, showed that those weighing ≥120 kg and treated with apixaban had a higher, but statistically insignificant, VTE recurrence rate than those weighing <120 kg [13].

A retrospective cohort study by Barakat AF et al. demonstrated that DOAC patients in all BMI categories, including underweight and obese, had a significantly lower risk of both types of strokes. Notably, this study also revealed that DOAC patients with BMIs of 40 or higher had a 25% and 50% reduced risk of ischemic and hemorrhagic stroke, respectively. It is worth noting that this study used ICD-9 and ICD-10 codes to identify outcomes in a large US hospital system, whereas we derived our outcomes from a thorough review of each patient’s electronic medical record [14].

A retrospective clinical study by Kushnir M. et al. highlighted that morbidly obese DOAC patients, including those with a BMI over 50 kg/m², exhibited stroke, recurrent venous thromboembolism, and major bleeding rates comparable to those on warfarin [15].

The recent ISTH SSC Subcommittee and Expert Consensus Panel updates on DOACs in obese VTE patients highlighted the need for clearer therapeutic targets, noting that efficacy gaps persist, especially in severe obesity and post-bariatric surgery. Indeed, the subcommittee looks mostly at VTE treatment and prevention but does not include the atrial fibrillation group [16]. Despite the 2021 removal of BMI limitations, which had previously advised against DOAC use in patients with severe obesity, hesitancy remains among healthcare providers due to limited data on efficacy and safety in high-BMI populations. These findings underscore the importance of our real-world investigation into DOAC outcomes in obese VTE and AF patients, broadening insights into treatment efficacy in high-risk populations [17].

### 4.2. Bleeding Literature of Doacs vs. Warfarin in Obesity

Coons et al. found no significant discrepancy in major bleeding incidence rates between DOACs and warfarin, aligning with our results [11]. Several other studies also noted that DOACs had a lower association with bleeding events [11,14]. Barakat et al. observed a strong trend towards lower risks of bleeding events with DOAC usage, although this did not achieve statistical significance. Their research also revealed a reduction in bleeding risk by about 60% in morbidly obese patients on DOACs [14]. Interestingly, genitourinary bleeding was seen more commonly among DOAC-treated patients, whereas gastrointestinal bleeding occurred more frequently in warfarin-treated patients [11]. Although this distribution of bleeding aligns with our study, the difference in gastrointestinal bleeding risk remains non-significant, as reported in past studies [18,19]. An observational meta-analysis involving 1,332,956 non-valvular AF patients found no significant difference in gastrointestinal bleeding rates between rivaroxaban and warfarin [18]. Additionally, a review of randomized controlled trials (RCTs), retrospective database studies, and large-scale prospective cohort studies reported no substantial difference in major gastrointestinal bleeding risk between DOACs and warfarin, although they noted that DOAC-treated patients experienced less severe gastrointestinal bleeding and required less intensive management [19]. Our findings are consistent with a recent meta-analysis by Karakasis et al., which showed that DOACs are equally effective as warfarin and offer a better safety profile in obese patients (BMI ≥ 30 kg/m^2^), including a lower risk of bleeding. In addition to the findings from the meta-analysis, our study adds real-world data and employs propensity score matching to adjust for baseline characteristics that further support the use of DOACs in obese patients [20].

### 4.3. Mortality Literature of Doacs vs. Warfarin in Obesity

Our results showed a statistically significant reduction in mortality risk in AF patients treated with DOACs, a finding mirrored by Barakat et al. They found that all BMI groups, except for the underweight, experienced lower mortality rates with DOACs compared to warfarin, with a 34% reduction in all-cause mortality in patients with a BMI of 40 or above [14]. Law et al. also found an association between lower all-cause mortality rates in females and DOAC use compared to warfarin, although the BMI of the population was unspecified [21]. This association persisted but changed to statistically insignificant results after propensity matching. The superior efficacy of DOACs over warfarin remains to be explained. Some studies suggest that warfarin-treated patients may be less compliant, while others posit that warfarin’s inconsistent therapeutic range could affect its safety and effectiveness. These factors were not measured in our study, but the consistent link between DOACs and less adverse outcomes remains clinically significant.

### 4.4. Strengths and Limitation

Our study boasts several noteworthy strengths. First, we utilized the propensity-matching score to address possible confounding factors, thus heightening the credibility of our results. Second, our observational study provided the opportunity to examine the efficacy and safety of apixaban within a real-world context, offering useful insights. Lastly, our findings can influence clinical decisions and enhance patient outcomes, as they present evidence of apixaban efficacy and safety within a population that has not been thoroughly studied.

As the study is a retrospective observational study, there was a selection bias, which we addressed by adjusting for potential confounders. Another limitation is the reliance on electronic medical records, which have only been available in our center since 2015, resulting in a short follow-up period. The findings are limited by the focus of this treatment study on apixaban compared to warfarin. Incomplete data extraction limits our findings regarding the report of the bleeding risk score associated with AF patients like HAS-BLED or AREIA. Finally, it should be noted that we did not account for possible medication switchovers during the study period.

## 5. Conclusions

This study concluded that apixaban was associated with similar efficacy and safety compared to warfarin in obese patients. There is significantly lower all-cause mortality in high-BMI AF patients with apixaban compared to warfarin. This suggests that apixaban may be a safer alternative. More research is needed to endorse these findings and develop guidelines for DOAC use in this demographic. In essence, this study provides useful insight into DOACs in obese AF patients, aiding clinicians in making informed decisions about anticoagulation therapy.

## Figures and Tables

**Figure 1 biomedicines-13-00490-f001:**
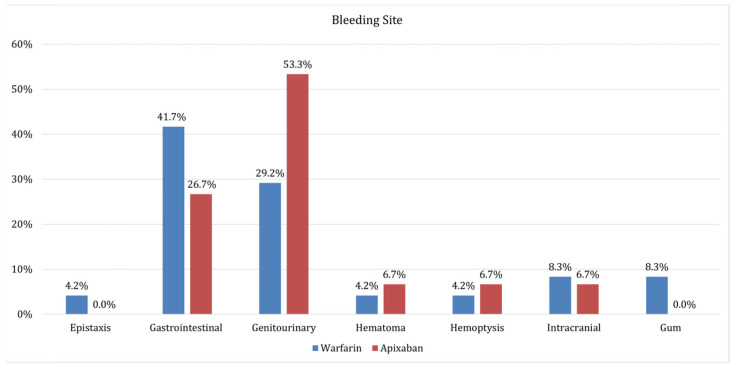
The prevalence of the bleeding on different sites.

**Figure 2 biomedicines-13-00490-f002:**
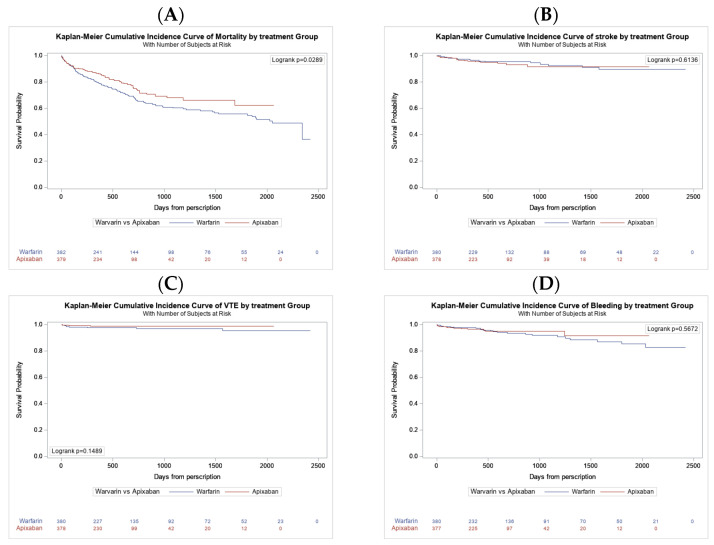
Kaplan–Meier cumulative incidence curves by treatment groups. (**A**) Mortality. (**B**) Stroke. (**C**) Venous thromboembolism (VTE). (**D**) Bleeding. The lines blue (warfarin) and red (apixaban) represent number of subjects at risk for each arm of the study population.

**Table 1 biomedicines-13-00490-t001:** Baseline characteristics pre- and post-matching for the study population. BMI: body mass index, CV: cardiovascular disease, MI: myocardial infarction, HF: heart failure, VTE: venous thromboembolism, PVD: peripheral Vascular disease, CKD: chronic kidney disease, CABG: coronary artery bypass grafting, Hx: history, CHA_2_DS_2_VASc: congestive heart failure; hypertension; age ≥ 75 years (doubled); diabetes mellitus; prior Stroke, TIA, or thromboembolism (doubled); vascular disease; age 65 to 74 years; and sex category. ^§^ Data represented as the median and interquartile (IQs). ° Data presented as the mean and standard deviation (SD).

Variable	Pre-Matching			*p* Value	Variable	Post-Matching			*p* Value
	Total 876	Warfarin 414	Apixaban 462			Total 772	Warfarin 386	Apixaban 386	
Age ^§^ (years)	77.0 (67.0, 85.0)	78.0 (67.0, 86.0)	76.0 (67.0, 83.0)	0.037	Age ^§^ (years)	78.0 (67.0, 86.0)	76.0 (67.0, 84.0)	77.0 (67.0, 85.0)	0.169
Gender (Female)	575 (65.6%)	260 (62.8%)	315 (68.2%)	0.094	Gender (Female)	242 (2.7%)	250 (64.8%)	492 (63.7%)	0.549
BMI ^§^	33.3 (31.0, 37.3)	32.9 (30.9, 36.6)	33.8 (31.2, 38.1)	0.018	BMI ^§^	32.8 (30.8, 36.5)	33.1 (31.1, 37.6)	32.9 (30.9, 36.9)	0.120
30 to <35	544 (62.1%)	266 (64.3%)	278 (60.2%)	0.238	30 to <35	252 (65.3%)	243 (63.0%)	495 (64.1%)	0.764
35 to <40	188 (21.5%)	89 (21.5%)	99 (21.4%)		35 to <40	81 (21.0%)	84 (21.8%)	165 (21.4%)	
≥40	144 (16.4%)	59 (14.3%)	85 (18.4%)		≥40	53 (13.7%)	59 (15.3%)	112 (14.5%)	
CV Risk Factors					CV Risk Factors				
Hypertension	698 (79.7%)	319 (77.1%)	379 (82.0%)	0.067	Hypertension	299 (77.5%)	306 (79.3%)	605 (78.4%)	0.541
Diabetes	578 (66.0%)	275 (66.4%)	303 (65.6%)	0.793	Diabetes	261 (67.6%)	247 (64.0%)	508 (65.8%)	0.288
Dyslipidemia	425 (48.5%)	184 (44.4%)	241 (52.2%)	0.023	Dyslipidemia	171 (44.3%)	197 (51.0%)	368 (47.7%)	0.061
Previous CV History					Previous CV history				
Previous MI	298 (34.0%)	135 (32.6%)	163 (35.3%)	0.405	Previous MI	129 (33.4%)	122 (31.6%)	251 (32.5%)	0.591
Hx of HF	471 (53.8%)	211 (51.0%)	260 (56.3%)	0.116	Hx of HF	195 (50.5%)	218 (56.5%)	413 (53.5%)	0.097
Hx of VTE	47 (5.4%)	31 (7.5%)	16 (3.5%)	0.008	Hx of VTE	10 (2.6%)	4 (1.0%)	14 (1.8%)	0.106
Hx of Stroke	177 (20.2%)	92 (22.2%)	85 (18.4%)	0.159	Hx of Stroke	10 (2.6%)	4 (1.0%)	14 (1.8%)	0.106
Hx of PVD	28 (3.2%)	15 (3.6%)	13 (2.8%)	0.497	Hx of PVD	14 (3.6%)	11 (2.8%)	25 (3.2%)	0.541
Hx of CKD	247 (28.2%)	127 (30.7%)	120 (26.0%)	0.123	Hx of CKD	118 (30.6%)	106 (27.5%)	224 (29.0%)	0.341
Hx of Cancer	100 (11.4%)	39 (9.4%)	61 (13.2%)	0.079	Hx of Cancer	37 (9.6%)	49 (12.7%)	86 (11.1%)	0.170
Hx of Anemia	101 (11.5%)	45 (10.9%)	56 (12.1%)	0.563	Hx of Anemia	42 (10.9%)	44 (11.4%)	86 (11.1%)	0.819
Previous Cath	168 (19.2%)	79 (19.1%)	89 (19.3%)	0.946	Previous Cath	77 (19.9%)	54 (14.0%)	131 (17.0%)	0.027
Previous CABG	10 (1.1%)	7 (1.7%)	3 (0.6%)	0.205	Previous CABG	7 (1.8%)	1 (0.3%)	8 (1.0%)	0.069
Additional Medications					Additional Medications				
Aspirin	565 (64.5%)	259 (62.6%)	306 (66.2%)	0.257	Aspirin	245 (63.5%)	242 (62.7%)	487 (63.1%)	0.823
Clopidogrel	227 (25.9%)	81 (19.6%)	146 (31.6%)	<0.001	Clopidogrel	81 (21.0%)	81 (21.0%)	162 (21.0%)	1.000
Calculated Risk Score					Calculated Risk Score				
CHA_2_DS_2_VASc °	4.9 ± 1.97	4.9 ± 1.94	5.0 ± 2.01	0.977	CHA_2_DS_2_VASc °	4.9 ± 1.90	4.8 ± 1.98	4.9 ± 1.94	0.454

**Table 2 biomedicines-13-00490-t002:** Outcomes of the study population pre- and post-matching. VTE: venous thromboembolism, ISTH: International Committee on Thrombosis and Haemostasis.

Variable	Pre-Matching			*p* Value	Variable	Post-Matching			*p* Value
	Total 876	Warfarin 414	Apixaban 462			Total 772	Warfarin 386	Apixaban 386	
Stroke	41 (4.8%)	21 (4.5%)	24 (4.6%)	0.842	Stroke	36 (4.7%)	18 (4.7%)	18 (4.7%)	1.000
VTE	15 (1.7%)	10 (2.4%)	5 (1.1%)	0.129	VTE	14 (1.8%)	10 (2.6%)	4 (1.0%)	0.107
Bleeding	47 (5.4%)	26 (6.3%)	21 (4.5%)	0.255	Bleeding	39 (5.1%)	24 (6.2%)	15 (3.9%)	0.139
ISTH (major)	11 (23.4%)	6 (23.1%)	5 (23.8%)	1.000	ISTH (major)	9 (23.1%)	6 (25.0%)	3 (20.0%)	1.000
Transfusion (>2 units)	7 (14.9%)	3 (11.5%)	4 (19.0%)	0.684	Transfusion (>2 units)	6 (15.4%)	3 (12.5%)	3 (20.0%)	0.658
Death	234 (26.7%)	139 (33.6%)	95 (20.6%)	<0.001	Death	207 (26.8%)	130 (33.7%)	77 (19.7%)	<0.001

**Table 3 biomedicines-13-00490-t003:** Cox model predicting all-cause mortality. * Risk factors included chronic kidney disease, cancer, hypertension, and a previous history of venous thromboembolism.

	Hazard Ratio	95% Confidence Interval	*p* Value
Apixaban vs. Warfarin			
Model 1: unadjusted	0.728	(0.55, 0.97)	0.030
Model 2: adjusted for age and gender	0.770	(0.58, 1.03)	0.076
Model 3: adjusted for model 2 + risk factors *	0.728	(0.55, 0.97)	0.032
Model 4A: adjusted for model 3 + BMI categories	0.729	(0.55, 0.97)	0.032
Model 4A: adjusted for model 3 + BMI continuous	0.727	(0.54, 0.97)	0.031

## Data Availability

The data supporting the reported results are not publicly available due to privacy and ethical restrictions but can be made available upon reasonable request to the corresponding author.

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
