# Peer review of "Real-World Efficacy and Safety of Apixaban vs. Warfarin in Obese Atrial Fibrillation Patients: Propensity Matching Analysis"

_biomedicines, 2025, doi:10.3390/biomedicines13020490_

Round 1

Reviewer 1 Report

Comments and Suggestions for Authors

per attached file

Author Response

  1. Science:
    1. The authors report the usage of 3 distinct DOACs: apixaban, dabigatran, or rivaroxaban. However, 92.5% of these DOAC patients were given apixaban, for a total of 480 individuals. It is not clear why the authors insisted on including in this study, the relatively small number of patients on dabigatran or rivaroxaban. Clearly, all other things being equal, any significant differences should be due to the effects of apixaban treatments. This would be the case unless the other two treatments had some unexpectedly outsized effects. Recommendation: Remove the dabigatran and rivaroxaban associated data and focus the analysis on the comparison between apixaban and warfarin treatments in matched patients.

We thank the reviewer for this unique and critical comment about empowering the study's generalizability by reporting the apixaban only. Also, besides this comment, we have excluded the VTE patients and kept AF-only patients in the primary analysis. The overall results showed significant changes. We have updated the manuscript, tables, and figures accordingly.

  1. Presentation: This review is well written, with only a few minor instances that can be improved.
    1. 24: Acronym 'CHA2DS2-VASc', elsewhere used in the form 'CHA2DS2VASc', has not been defined.
    2. 50: Change "direct thrombin/factor IIa" to "direct thrombin".

Done

  1. 64: Change "30kg/m2" to "30kg/m2"

Done

  1. 112: Change "generated as follow:" to "generated as follows:"

Done

  1. 114: Change "continuous, risk" to "continuous, and risk"

Done

  1. 178: Change “thromboembolism” to “thromboembolism).”

Done

  1. 271: Change "associated similar" to "associated with similar"

Done

  1. Table 1: Following the header "Previous CV history" there is no need to keep repeating either 'previous' or 'history of' for each diagnosis.

Done

  1. Table 1: Acronyms were listed well. However, they excluded an explanation for 'CHA2DS2VASc'

Done

  1. Table 2: Last row: Change "Death 51 (26.2%)" to "Death 151 (26.2%)"

These values had been updated.

Reviewer 2 Report

Comments and Suggestions for Authors

I find the article interesting and well written. I have only few concerns.

Firstly, since there is recent literature about this topic, I think that the authors might better discuss why their work could be of helping for physicians. In a recent systematic review Karakasis et al. found a notable between-study heterogeneity in participants’ baseline characteristics. So the authors could stress the value of the propensity score matching technique use.

Secondly, I don’t understand why the authors excluded patients with severe heart failure with an ejection fraction below 30% and why they didn’t included patients on therapy with edoxaban.

Lastly, about 92% of the analyzed patients were on therapy with apixaban and 94% were anticoagulated due to atrial fibrillation. Is it correct to generalize the results to atrial fibrillation and venous thromboembolism obese patients? And to DOACs? Or it is more correct that the authors present their data about atrial fibrillation obese patients on therapy with apixaban?

Specific comments:

Abstract:

Please modify: “Methods: A brief description of the main methods or treatments applied. This can include any relevant preregistration or specimen information.”

I think that a verb lacks in the following sentence: “The incidence of stroke (4.0% vs. 4.9%, p=0.507), venous 26 thromboembolism (VTE) (1.9% vs. 3.1%, p=0.269), and a trend to lower bleeding events (4.2% vs. 27 6.8%, p=0.099).”

Materials and Methods: No comments.

Results:

At page 4 on Table (1) I suggest that the values inside the brackets be on the same line.

At page 4 on lines 145-153 and at page 6 on lines 160-170, I think the authors didn’t want to use italics.

Discussion:

At page 7 on line 188: “PSM” was used here for the first time, so the authors should explain it.

At page 8 on lines 210-213: “The recent ISTH SSC Subcommittee and Expert Consensus Panel updates on DOACs in 210 obese VTE patients highlighted the need for clearer therapeutic targets, noting that efficacy gaps persist, especially in severe obesity and post-bariatric surgery. Indeed, it looks 212 mostly at VTE treatment and prevention but does not include the atrial fibrillation group.” I suggest to add “[15]” at the end of the sentence.

At page 9 on lines 248-250: “Law et al. also found an association between lower all-cause mortal- 248 ity rates in females and DOAC use compared to warfarin, although the BMI of the population was unspecified [16].” I think that the reference is not correct.

At page 9 on lines 267-268, I suggest to eliminate the following sentence: “Nevertheless, this limitation offers an opportunity for further research on other types of DOACs.”

Conclusions:

At page 9 on line 272 and on line 276 I suggest to modify “in AF patients” with “in AF and venous thromboembolism”. See also above (my last general comment).

Author Response

I find the article interesting and well-written. I have only a few concerns.

  1. Firstly, since there is recent literature about this topic, I think that the authors might better discuss why their work could be of helping for physicians. In a recent systematic review Karakasis et al. found a notable between-study heterogeneity in participants' baseline characteristics. So, the authors could stress the value of the propensity score matching technique use.

Thank you for the reviewer. We have updated the first paragraph of the discussion as following: “Furthermore, since feasibility and ethics are always an issue for randomized clinical trials, the implementation of statistical methods significantly reduces the impact of existing confounders. Propensity-matching score, as one of the statistical solutions, estimates the relevant clinical effects adjusted for given confounders. Therefore, to overcome the literature and design limitations, we utilized a propensity-matching score to prove the extent of efficacy and safety in different unstudied populations.”

  1. Secondly, I don't understand why the authors excluded patients with severe heart failure with an ejection fraction below 30% and why they didn't included patients on therapy with edoxaban.

Thank you for the reviewer for highlighting this highly important point. We think that patients with severe heart failure of ejection fraction lower than 30% are at high risk of developing LV clot irrespective of presence of atrial fibrillation.  

  1. Lastly, about 92% of the analyzed patients were on therapy with apixaban and 94% were anticoagulated due to atrial fibrillation. Is it correct to generalize the results to atrial fibrillation and venous thromboembolism obese patients? And to DOACs? Or it is more correct that the authors present their data about atrial fibrillation obese patients on therapy with apixaban?

 We thank the reviewer for this unique and critical comment about empowering the study's generalizability by reporting the apixaban only. Also, besides this comment, we have excluded the VTE patients and kept AF-only patients in the primary analysis. The overall results showed significant changes. We have updated the manuscript, tables, and figures accordingly.

Specific comments:

  1. Abstract:
    1. Please modify: "Methods: A brief description of the main methods or treatments applied. This can include any relevant preregistration or specimen information."

Thank you for the reviewer comment. We have updated the methods section of the abstract as following: “A retrospective cohort study examined consecutive AF patients with a BMI of ≥ 30 kg/m2 treated with apixaban or warfarin. Patients was started on these medications between January 2015 to December 2021. Efficacy outcomes included ischemic stroke and venous thromboembolism (VTE) occurrences, while safety outcomes encompassed bleeding incidents and mortality rates. Out-comes were assessed following propensity score matching.”

  1. I think that a verb lacks in the following sentence: "The incidence of stroke (4.0% vs. 4.9%, p=0.507), venous 26 thromboembolism (VTE) (1.9% vs. 3.1%, p=0.269), and a trend to lower bleeding events (4.2% vs. 27 6.8%, p=0.099)."

Thank you for the reviewer comment. We have updated this section as following: “The incidences of stroke, venous thromboembolism (VTE) , and bleeding events were  (4.7% vs. 4.7%, p=1.000), (1.0% vs. 2.6%, p=0.107), and (3.9% vs. 6.2%, p=0.139), respectively.”

  1. Materials and Methods: No comments.

  1. Results:
    1. At page 4 on Table (1) I suggest that the values inside the brackets be on the same line.

Done

  1. At page 4 on lines 145-153 and at page 6 on lines 160-170, I think the authors didn't want to use italics.

Thank you for the reviewer. We agree with this point, sections 3.2 and 3.3 were updated.

  1. Discussion:
    1. At page 7 on line 188: "PSM" was used here for the first time, so the authors should explain it.

We agree with reviewer. This abbreviation was spelled out.

  1. At page 8 on lines 210-213: "The recent ISTH SSC Subcommittee and Expert Consensus Panel updates on DOACs in 210 obese VTE patients highlighted the need for clearer therapeutic targets, noting that efficacy gaps persist, especially in severe obesity and post-bariatric surgery. Indeed, it looks 212 mostly at VTE treatment and prevention but does not include the atrial fibrillation group." I suggest to add "[15]" at the end of the sentence.

The reference was updated.

  1. At page 9 on lines 248-250: "Law et al. also found an association between lower all-cause mortal- 248 ity rates in females and DOAC use compared to warfarin, although the BMI of the population was unspecified [16]." I think that the reference is not correct.

We agree with reviewer. The reference was updated on the text and the appropriate reference was added to the reference list.

  1. At page 9 on lines 267-268, I suggest to eliminate the following sentence: "Nevertheless, this limitation offers an opportunity for further research on other types of DOACs."

Thank you for the reviewer. We agree on the suggestion, the sentence was deleted.

  1. Conclusions:
    1. At page 9 on line 272 and on line 276 I suggest to modify "in AF patients" with "in AF and venous thromboembolism". See also above (my last general comment).

Based on the other reviewers’ comment, we updated the results by focusing the study on diagnosis of AF and patients received apixaban.

Reviewer 3 Report

Comments and Suggestions for Authors

Comments to the authors

Re: Manuscript Number: biomedicines-3394722; Title: Real-world efficacy and safety of DOACs vs warfarin in atrial fibrillation and venous thromboembolism obese patients: Propensity matching analysis.

Dr. Algethami and colleagues have completed a retrospective study to assess the real-world safety and effectiveness of DOACs compared to warfarin in treating atrial fibrillation (AF) and venous thromboembolism (VTE) in obese patients. This is a meaningful topic. However, there are still some following issues needing to be revised.

1.   On Abstract section, the Method is not clear.

2.   On Introduction section, the citations are not enough. Furthermore, please complement the relationship and international research progress between obesity and VTE.

3.   On Results section, please complement HAS-BLED score in Table 1. Furthermore, it may be better to describe 3.1 and 3.2 together in a subtitle.

4.   On Table 2 and Figure 1, every patient has different follow up duration, and it is not reasonable to compare them directly without a definite timepoint. Please compare them in different timepoint like 3-month, 1-year and 2-year outcomes. 

5.   On Table 3, how do you choose the risk factors in model 4?

6.   Some mistakes are existed in sentence expression, paper format and data presentation. Please check up the data and manuscript carefully.

Author Response

Re: Manuscript Number: biomedicines-3394722; Title: Real-world efficacy and safety of DOACs vs warfarin in atrial fibrillation and venous thromboembolism obese patients: Propensity matching analysis.

 Dr. Algethami and colleagues have completed a retrospective study to assess the real-world safety and effectiveness of DOACs compared to warfarin in treating atrial fibrillation (AF) and venous thromboembolism (VTE) in obese patients. This is a meaningful topic. However, there are still some following issues needing to be revised.

  1. On Abstract section, the Method is not clear.

Thank you for the reviewer comment. We have updated the methods section of the abstract as following: “A retrospective cohort study examined consecutive AF patients with a BMI of ≥ 30 kg/m2 treated with apixaban or warfarin. Patients was started on these medications between January 2015 to December 2021. Efficacy outcomes included ischemic stroke and venous thromboembolism (VTE) occurrences, while safety outcomes encompassed bleeding incidents and mortality rates. Out-comes were assessed following propensity score matching.”

  1. On Introduction section, the citations are not enough. Furthermore, please complement the relationship and international research progress between obesity and VTE.

Thank you for the reviewer’s comment; however, based on the other reviewers’ comment, we updated the results by focusing the study on diagnosis of AF and patients received apixaban. All patients who received any other DOAC except apixaban was excluded and the single indication for using apixaban was AF.

  1. On Results section, please complement HAS-BLED score in Table 1. Furthermore, it may be better to describe 3.1 and 3.2 together in a subtitle.

We thank the reviewer for this point. We used CHA2DS2-VASc score to propose the patients’ risk.

The main strength point is using statical method to overcome the weight of the confounder on the study outcomes. Therefore, we would like to show the original data at the baseline before propensity matching in separate from the post propensity for better demonstration. However, these two sections were lined side by side in table 1 for easy comparison.

  1. On Table 2 and Figure 1, every patient has different follow up duration, and it is not reasonable to compare them directly without a definite timepoint. Please compare them in different timepoint like 3-month, 1-year and 2-year outcomes. 

Thank you for the reviewer. We updated the results section with the following statement: “Additional figures show the event periodically available in the appendix (Sup Fig 1-(A-D)).” and we would like these following figures to be added to the appendix of manuscript.

  1. On Table 3, how do you choose the risk factors in model 4?

We thank the reviewer for the comment. We think that these risk factors have high association with the study outcome “all-cause mortality”

  1. Some mistakes are existed in sentence expression, paper format and data presentation. Please check up the data and manuscript carefully.

Thank you for highlighting this issue. The current version of the manuscript was carefully revised and all required changes were made.

Round 2

Reviewer 1 Report

Comments and Suggestions for Authors

Summary:
This is a revised report of a clinical outcomes study of obese patients, BMI ≥ 30, who were
treated for atrial fibrillation (AF) either by warfarin or by apixaban. This study reports a
significantly lower all-cause mortality after treatment with apixaban compared to warfarin
treatments. This is of substantial interest.
Feedback:
1) Science:
a) The authors have refocused their report on the comparison between warfarin and apixaban
treatments. By so doing, they have made the report clearer and easier to interpret.
2) Presentation: After substantial revision and textual refocusing, some new corrections are
needed. The following are some examples.
a) 23: Change “Patients was started” to “Patients were started”
b) 38: Change “which persisted to be significant after” to “which remained significant after”
c) 39-41: “Apixaban and warfarin have efficacy in preventing thromboembolism among
obese patients with atrial fibrillation; though, apixaban had significantly lower the all-cause
mortality.” This statement is misleading. For the first part of this statement to be true, the
comparison would be needed with a third control group receiving no treatment. However,
that is not presented here for obvious ethical reasons. Therefore, change sentence to:
“Apixaban is associated with a trend of reduced incidence of thromboembolism among
obese patients with atrial fibrillation, and significantly lower all-cause mortality.”
d) 43: Change “obese patients AF.” to “obese patients with AF.”
e) 140: Include full description for first usage of the “CHA2DS2-VASc score”.
f) 145: Change “Apixaban were administered” to “Apixaban was administered”
g) 180: Change “Safety and efficacy of DOAC” to “Safety and efficacy of Apixaban”
h) 189-190: Change “The lower mortality risk associated with apixaban persisted to be
significant after” to “The mortality risk associated with apixaban is significantly lower
even after”
i) Figure 2: Presentation of the survival probability curves could be more helpful. Instead of
starting at 0, it would be clearer if the y axes were started closer to the actual data.
j) 301-302: Change “The generalizability of our findings was limited, since our cohort was
received single DOAC apixaban.” to “The findings are limited by the focus of this
treatment study on apixaban compared to warfarin.”
k) 307: Change “that apixaban were associated” to “that apixaban was associated”

Author Response

  1. Science:
    1. The authors have refocused their report on the comparison between warfarin and apixaban treatments. By so doing, they have made the report clearer and easier to interpret.

We Thank the reviewer for highlighting the point of emphasizing the manuscript on reporting the apixaban rather than all DOACs, which made our work more interpretable.

  1. Presentation: After substantial revision and textual refocusing, some new corrections are

needed. The following are some examples.

  1. 23: Change "Patients was started" to "Patients were started"

Done

  1. 38: Change "which persisted to be significant after" to "which remained significant after"

Done

  1. 39-41: "Apixaban and warfarin have efficacy in preventing thromboembolism among obese patients with atrial fibrillation; though, apixaban had significantly lower the all-cause mortality." This statement is misleading. For the first part of this statement to be true, the comparison would be needed with a third control group receiving no treatment. However, that is not presented here for obvious ethical reasons. Therefore, change sentence to: "Apixaban is associated with a trend of reduced incidence of thromboembolism among obese patients with atrial fibrillation, and significantly lower all-cause mortality."

Done

  1. 43: Change "obese patients AF." to "obese patients with AF."

Done

  1. 140: Include full description for first usage of the "CHA2DS2-VASc score".

Done

  1. 145: Change "Apixaban were administered" to "Apixaban was administered"

Done

  1. 180: Change "Safety and efficacy of DOAC" to "Safety and efficacy of Apixaban"

Done

  1. 189-190: Change "The lower mortality risk associated with apixaban persisted to be significant after" to "The mortality risk associated with apixaban is significantly lower even after"

Done

  1. Figure 2: Presentation of the survival probability curves could be more helpful. Instead of starting at 0, it would be clearer if the y axes were started closer to the actual data.

Done

  1. 301-302: Change "The generalizability of our findings was limited, since our cohort was received single DOAC apixaban." to "The findings are limited by the focus of this treatment study on apixaban compared to warfarin."

Done

  1. 307: Change "that apixaban were associated" to "that apixaban was associated"

Done

Reviewer 3 Report

Comments and Suggestions for Authors

1.   About question 3 “On Results section, please complement HAS-BLED score in Table 1. Furthermore, it may be better to describe 3.1 and 3.2 together in a subtitle.”, CHA2DS2-VASc score is used as assessing the risk of ischemic stroke and HAS-BLED score is used as assessing the risk of bleeding, which are both quite important for AF patients. 

2.   About question 4 “On Table 2 and Figure 1, every patient has different follow up duration, and it is not reasonable to compare them directly without a definite timepoint. Please compare them in different timepoint like 3-month, 1-year and 2-year outcomes.”, please complement the P values of different groups in Sup Fig 1A-D. Furthermore, these figures could not be found in the text, and please submit them as supplementary materials instead of putting them in “author-response” form.

3.   About question 5 “On Table 3, how do you choose the risk factors in model 4?”, the authors should be rigorous when doing research instead of according to your thought. Risk factors usually come from previous studies or statistically significant variables from your research.

Author Response

  1. About question 3 “On Results section, please complement HAS-BLED score in Table 1. Furthermore, it may be better to describe 3.1 and 3.2 together in a subtitle.”, CHA2DS2-VASc score is used as assessing the risk of ischemic stroke and HAS-BLED score is used as assessing the risk of bleeding, which are both quite important for AF patients.

Thank you for the reviewer for the comment. We merge the two sections of 3.1 and 3.2 together in a single subtitle.

We agree with the reviewer’s comment about the value of reporting bleeding risk. Unfortunately, the data set we have lacks variables to allow us to calculate HAS-BLED score. Understating the high value of such score in the practice, we thought of calculating ATRIA score to overcome the previous limitation. Unfortunately, we faced the same issue of missing the data to calculate the score. Therefore, while we regretfully cant answer the reviewer comment, we have added this point to the limitations of the study “Incomplete data extraction limits our findings to report bleeding risk score associated with AF patients like HAS-BLED or AREIA.”

  1. About question 4 “On Table 2 and Figure 1, every patient has different follow up duration, and it is not reasonable to compare them directly without a definite timepoint. Please compare them in different timepoint like 3-month, 1-year and 2-year outcomes.”, please complement the P values of different groups in Sup Fig 1A-D. Furthermore, these figures could not be found in the text, and please submit them as supplementary materials instead of putting them in “author-response” form.

Thank you for the reviewer comment. We added a section to the manuscript “Supplementary materials” with all the four figures of different outcomes after update of p-values.

  1. About question 5 “On Table 3, how do you choose the risk factors in model 4?”, the authors should be rigorous when doing research instead of according to your thought. Risk factors usually come from previous

We thank the reviewer for their valuable comment. The model one was unadjusted, for model two we adjusted for age and gender, for model three we adjusted for Chronic Kidney Disease, Cancer, hypertension, previous history of venous thromboembolism in addition to age and gender. This is based on work by Vinogradova et al. in BMJ (doi: https://doi.org/10.1136/bmj.k2505) showing these as some of the confounding factors not traditionally included in the CHA2DS2-VASc score. While the model 4, we adjusted for BMI in addition to the factors in the previous models. The reference was added to the referencing list and update the list accordingly.

Round 3

Reviewer 3 Report

Comments and Suggestions for Authors

The authors have answered all my questions. No more questions

Author Response

comment: The authors have answered all my questions. No more questions

answer: thank you